

# Crystal structure correlations with the intrinsic thermodynamics of human carbonic anhydrase inhibitor binding

Alexey Smirnov[1], Asta Zubrienė[1], Elena Manakova[2], Saulius Gražulis[2] and Daumantas Matulis[1]

[1] Department of Biothermodynamics and Drug Design, Institute of Biotechnology, Vilnius University, Vilnius, Lithuania
[2] Department of Protein-DNA Interactions, Institute of Biotechnology, Vilnius University, Vilnius, Lithuania

## ABSTRACT

The structure-thermodynamics correlation analysis was performed for a series of fluorine- and chlorine-substituted benzenesulfonamide inhibitors binding to several human carbonic anhydrase (CA) isoforms. The total of 24 crystal structures of 16 inhibitors bound to isoforms CA I, CA II, CA XII, and CA XIII provided the structural information of selective recognition between a compound and CA isoform. The binding thermodynamics of all structures was determined by the analysis of binding-linked protonation events, yielding the intrinsic parameters, i.e., the enthalpy, entropy, and Gibbs energy of binding. Inhibitor binding was compared within structurally similar pairs that differ by *para-* or *meta-*substituents enabling to obtain the contributing energies of ligand fragments. The pairs were divided into two groups. First, *similar* binders—the pairs that keep the same orientation of the benzene ring exhibited classical hydrophobic effect, a less exothermic enthalpy and a more favorable entropy upon addition of the hydrophobic fragments. Second, *dissimilar* binders—the pairs of binders that demonstrated altered positions of the benzene rings exhibited the non-classical hydrophobic effect, a more favorable enthalpy and variable entropy contribution. A deeper understanding of the energies contributing to the protein-ligand recognition should lead toward the eventual goal of rational drug design where chemical structures of ligands could be designed based on the target protein structure.

## INTRODUCTION

Low molecular weight compound recognition by a target protein is still rather poorly understood hindering efficient rational design of novel molecules that would tightly and specifically bind selectively to the target protein and could be developed into an efficient and non-toxic drug that does not bind and influence the function of other vital proteins. Protein-compound complexes could be characterized structurally and thermodynamically. Correlations between these approaches is the main focus of this manuscript.

Corresponding author
Daumantas Matulis, matulis@ibt.lt, daumantas.matulis@bti.vu.lt

Carbonic anhydrases (CAs, EC 4.2.1.1) are ubiquitous metallo-enzymes which catalyze the hydration reaction of carbon dioxide into bicarbonate. Human CAs belong to the $\alpha$-family comprised of fifteen isozymes which have different cellular localization, distribution, kinetic properties, expression levels, and oligomerization state (*Sly & Hu, 1995*; *Supuran, 2008*; *Supuran, 2015*; *Alterio et al., 2012*). CAs are involved in many physiological and pathological processes. Twelve of the isozymes (isoforms) contain Zn(II) metal in the active site coordinated by three histidines, while remaining three CA isozymes do not possess any catalytic activity because they do not have Zn(II) in the active site due to the absence of one or more His residues (*Aspatwar et al., 2014*) needed for the coordination of the metal.

The 12 human CA isoforms have a high degree of conservative residues between themselves. The amino acids which may form contacts with the inhibitor are nearly identical, especially in the deeper parts of the conical cavity-like active site (*Pinard, Mahon & McKenna, 2015*). These minor structural differences cause some inhibitors to exhibit million-fold selectivities between isoforms—a phenomenon that is difficult to explain and predict from the inhibitor chemical structure. CAs have been used as models of protein-ligand binding for a long time (*Krishnamurthy et al., 2008*; *Supuran, 2015*). There are over 500 CA structures with various bound ligands in the PDB database. However, most structures are available for CA II while other isoforms are represented sparsely, therefore limiting our ability to understand the structural basis of inhibitor selectivity.

A large number of studies have been published where novel series of compounds have been described. Compounds of various seemingly non-related chemical classes appear to be good inhibitors of CAs (*Lomelino & McKenna, 2016*; *Supuran, 2017*). Thus, aromatic sulfonamides are not the only available inhibitors, but the oldest group of inhibitors with the largest variation of chemical structures tested towards most CA isoforms. The mechanism of sulfonamide binding is rather well understood. The binding reaction is highly pH-dependent due to the linked protonation-deprotonation reactions of sulfonamide and the water molecule bound to the Zn(II) in the active site (*Taylor, King & Burgen, 1970*; *Pilipuitytė & Matulis, 2015*). It is important to dissect those linked reactions and calculate the intrinsic parameters that would be independent of the buffer and pH (*Krishnamurthy et al., 2008*; *Baranauskienė & Matulis, 2012*; *Jogaitė et al., 2013*; *Morkūnaitė et al., 2015*). Here we correlate the thermodynamics of binding with the structures of a series of fluorinated and chlorinated benzenesulfonamides that have been previously described *Kišonaitė et al. (2014)*, *Zubrienė et al. (2015)* and *Zubrienė et al. (2016)*.

The thermodynamics of ligand binding depends on many factors, including: (a) classical interaction such as hydrogen bonds, salt bridges, direct Van-der-Waals contacts, and hydrophobic interactions; (b) solvent-related effects (solvation, desolvation, the capture or release of water molecules) before and upon binding process; (c) conformational mobility and conformational changes of the ligand and the protein active site residues upon binding; (d) shape and exposed surface topography of the active site. Much of the thermodynamic data has been determined by isothermal titration calorimetry (ITC) technique that has been extensively reviewed (*Pethica, 2015*; *Falconer, 2016*) with the suggestions for the interpretation of ITC data.

The role of non-covalent interactions and reorganization of water molecules around the ligand and in the active site for the application of thermodynamics for drug design has been discussed in *Klebe (2015)*. There is an interesting suggestion that water molecules may have minor impact to the free energy of ligand binding, but relatively large effects to the changes of entropy and enthalpy. In the review by *Geschwindner, Ulander & Johansson (2015)*, the authors also remind that two standard constituents ($\Delta H$ and $\Delta S$) of the Gibbs energy of ligand-protein binding comprise in reality a mixture of many contributions which are discussed here.

In the current review on the ligand binding thermodynamics (*Claveria-Gimeno et al., 2017*) the binding enthalpy change is split into contributions: $\Delta H^{\text{observed}} = \Delta H_{\text{interactions}} + \Delta H_{\text{desolvation}} + \Delta H_{\text{conformational}} + \Delta H_{\text{exchange}}$, where $\Delta H_{\text{interactions}}$ is the enthalpy gain associated with formation of non-covalent interactions between binding partners, $\Delta H_{\text{conformational}}$ is the enthalpy gain associated with the conformational changes of ligand and protein, $\Delta H_{\text{desolvation}}$ is the enthalpy gain associated with the desolvation processes of the binding partners, $\Delta H_{\text{exchange}}$ is the enthalpy gain associated with additional impact from interactions between ligand or protein and another buffer compounds, such as protons, small molecules. The pH influence to the binding process can be determined and removed from $\Delta H_{\text{exchange}}$. The $\Delta H_{\text{desolvation}}$, $\Delta H_{\text{conformational}}$ and $\Delta H_{\text{interactions}}$ reflect enthalpies of binding steps: (1) desolvation of binding partners, (2) conformation changes of the partners upon binding, and (3) formation of non-covalent interactions between binding partners. The binding entropy change (multiplied by the absolute temperature) also can be split in a similar way: $-T\Delta S^{\text{observed}} = -T\Delta S_{\text{desolvation}} - T\Delta S_{\text{conformational}} - T\Delta S_{\text{exchange}} - T\Delta S_{\text{roto-translational}} - T\Delta S_{\text{vibrational}}$, where new members $T\Delta S_{\text{roto-translational}}$ and $T\Delta S_{\text{vibrational}}$ are roto-translational (associated with degree of translational and rotational freedom) and vibrational (associated with covalent bonds) entropies. The authors propose to focus on desolvation and conformational entropies. For example, the conformational entropy is usually unfavorable due to reduction in mobility of binding partners.

The changes in enthalpy upon binding could provide an additional information for selecting compounds in the lead discovery and optimization. The enthalpy-driven compounds often have been considered to have better probability to become a drug (*Ladbury, Klebe & Freire, 2010*). The favorable binding enthalpy can indicate the better geometry of hydrogen bonds, Van-der-Waals, and other interactions between the ligand and the target protein. The enthalpy gain is necessary but not sufficient for an affinity gain due to compensating entropic effects (*Freire, 2008*).

Whitesides with co-workers provided a comprehensive crystallographic analysis of a series of homologous compounds bound to CA II that was used in several studies of the hydrophobic effect. In the review by *Snyder et al. (2013)*, the authors take note of the influence of water dynamics during ligand binding on the known effect of enthalpy/entropy compensation. Water molecules in the binding pocket and around the ligand are the critical component in the process of protein-ligand binding. Another direction of the research was the use of CA II as a tool for the investigation of Hoffmeister series of ions. The structure of water molecules around the ligand have significant influence to the binding behavior of

a molecule to various surfaces. *Fox et al. (2015)* showed that the binding affinity of anions to the active site of CA II correlates inversely with their affinity for water (free energy of hydration), the poorly hydrated anions associate more effectively with cation (Zn(II)) and non-polar regions of CA II.

In the article by *Winquist et al., (2013)*, the authors showed the complex view of the binding processes by analyzing the binding thermodynamics, kinetics and crystal structures showing that the additional hydrogen bonding is not necessarily reflected in the enthalpic gain. Moreover, the minor changes in the structure of compound can influence in an unpredictable way the kinetic and thermodynamic signatures of the binding reaction due to the small structural rearrangements within the protein binding site and subtle changes in the relative orientation of the compounds.

Davis and colleagues studied the water structural transformation at molecular interfaces using spectroscopic observations. They reported that the properties of water molecules bound in the hydrophobic hydration shell differ from the bulk water in the degree of order in tetrahedrally hydrogen-bonded molecules and in the number of so-called dangling water (defects in water's tetrahedral H-bond network). The hydration shell has a greater degree of tetrahedral order and reduced population of the weakest hydrogen bonds at low temperatures (*Davis et al., 2012*). The number of dangling waters, which depends on the temperature, intramolecular charge delocalization, size of hydrophobic domains, also changes near the hydrophobic surface (*Davis et al., 2013*; *Davis et al., 2015*).

To better understand the binding and recognition process between a protein and a low-molecular-weight chemical compound, a system of closely related proteins and structurally similar compounds is necessary. Carbonic anhydrase isoforms comprise a nice group of proteins that are recognized by aromatic sulfonamides because the sulfonamide group binds to the active site Zn(II) cation. Because of this recognition, aromatic sulfonamides bind stoichiometrically and specifically to CAs. However, minor differences in the CA isoforms and differences in the tail part of the sulfonamides cause significant difference in the energies of binding. Therefore, the system is highly suitable for the detailed investigation of structure-thermodynamics relationship of the recognition process.

## MATERIALS AND METHODS

### Protein preparation
Recombinant human carbonic anhydrases isoforms were expressed and purified by affinity chromatography as described previously (CA I in *Čapkauskaitė et al. (2013)*, CA II in *Čapkauskaitė et al. (2012)*, CA XIII in *Sūdžius et al. (2010)*, and CA XII in *Jogaitė et al. (2013)*.

### Chemistry
The synthesis, chemical structural characterization, and the purity of 4-substituted-2,3,5,6-tetrafluorobenzenesulfonamides **1–5** and **10** has been described in *Dudutienė et al. (2013)*, chlorinated benzenesulfonamides **6–9** in *Čapkauskaitė et al. (2013)* and 3,4-disubstituted-2,5,6-trifluorobenzenesulfonamides **11–16** in *Dudutienė et al. (2014)* and *Dudutienė et al. (2015)*. The purity of all compounds was >95%, as verified by HPLC.

**Table 1** **The crystallization conditions used to grow the crystals obtained in this study.** The crystals were grown in the thermostatic room at 19 °C using vapor diffusion method of protein crystallization (sitting drop format). The crystallization plates were Cryschem Plate (24 well sitting drop). The volume of crystallization buffer was 0,5 mL in the well of crystallization plate. The mixed volumes of starting drop are listed in table below. The crystallization buffers were homemade using chemicals of highest available purity purchased from Carl Roth GmbH (Karlsruhe, Germany) and Sigma Aldrich (Munich, Germany) and optimized for crystallization of every presented CA isoform.

| Isoform-ligand | Crystallization buffer | PDB ID | Sitting drop |
|---|---|---|---|
| CA II-**4** (VD10-49) | 2.03 M sodium malonate (pH 7.5)[a] | 5LLH | 6 μL of 20 mg mL$^{-1}$ CA II protein solution and 6 μL of crystallization buffer |
| CA II-**13** (VD11-26) | 0.1 M sodium bicine (pH 9) (Roth #9162.3), 0.2 M ammonium sulfate (Sigma-Aldrich #31119) and 2 M sodium malonate (pH 7) | 5LLC | 5 μL of 40 mg mL$^{-1}$ CA II protein solution and 5 μL of crystallization buffer |
| CA II-**2** (VD10-51) | 2.5 M sodium malonate (pH 7.5) and 0.075 M sodium bicine (pH 9) | 5LLE | 6 μL of 60 mg mL$^{-1}$ CA II protein solution and 6 μL of crystallization buffer |
| CA II-**1** (VD12-10) | 0.1 M sodium bicine (pH 9), 0.2 M ammonium sulfate and 2 M sodium malonate (pH 7) | 5LLG | 4 μL of 50 mg mL$^{-1}$ CA II protein solution and 4 μL of crystallization buffer |
| CA XII-**10** (VD10-35) | 0.1 M ammonium citrate (pH 7) (Roth #9488.2), 0.2 M ammonium sulfate and 26% PEG4000 (Roth #0156.1) | 5MSB | 3 μL of 19.7 mg mL$^{-1}$ CA XII protein solution and 3 μL of crystallization buffer |
| CA XII-**16** (VD12-23) | 0.1 M ammonium citrate (pH 7), 0.2 M ammonium sulfate and 26% PEG4000 | 5LLP | 3 μL of 19.7 mg mL$^{-1}$ CA XII protein solution and 3 μL of crystallization buffer |
| CA XII-**12** (VD12-34) | 0.1 M ammonium citrate (pH 7), 0.2 M ammonium sulfate and 26% PEG4000 | 5LLO | 3 μL of 24.4 mg mL$^{-1}$ CA XII protein solution and 3 μL of crystallization buffer |

**Notes.**
[a]Sodium Malonate solution was prepared by titration of 2.7 M Malonic Acid ("Fluka" #63290) with NaOH powder ("Roth" #9356.2).

### X-ray crystallography

Crystallization conditions of seven crystal structures are listed in Table 1. All crystallization experiments were carried out in Cryschem sitting drop plates ("Hampton Research" Cat. Nr. HR3-160) at 18–20 °C. Crystals of CA II and CA XII were soaked with 0.5 mM solution of the ligand prepared in the crystallization buffer. The quality of the electron density of the compounds is presented in Fig. 1. The crystal structures were solved by molecular replacement using MOLREP software (*Vagin & Teplyakov, 1997*). Initial models were the following: 3HLJ for CA II, 2NNO for CA XIII, and 1JD0 for CA XII. The 3D models of compounds were generated using AVOGADRO (*Hanwell et al., 2012*). The library files which contain complete chemical and geometric descriptions of compounds were created using LIBCHECK (*Vagin et al., 2004*). Model building and refinement were made using COOT (*Emsley & Cowtan, 2004*) and REFMAC (*Murshudov, Vagin & Dodson, 1997*) programs, respectively. The graphics were generated using MOLSCRIPT (*Kraulis, 1991*), BOBSCRIPT (*Esnouf, 1999*), RASTER3D (*Merritt & Bacon, 1997*) and PYMOL (*Schrödinger, 2010*) programs. Coordinates and structure factors have been deposited to the RCSB Protein Data Bank (PDB). PDB access codes are listed in Table 2.

### Conformational changes

The root-mean-square deviations (RMSD) between CA-inhibitor complexes and apo proteins without ligand were calculated by Perl script for protein backbone atoms of chains A of crystal structures. The RMSD of the CA II complexes in comparison with apoCA II was 0.23 ± 0.04 Å (compound **7**—0.30 Å, **3**—0.22 Å, **4**—0.29 Å, **10**—0.18 Å, **8**—0.19 Å,
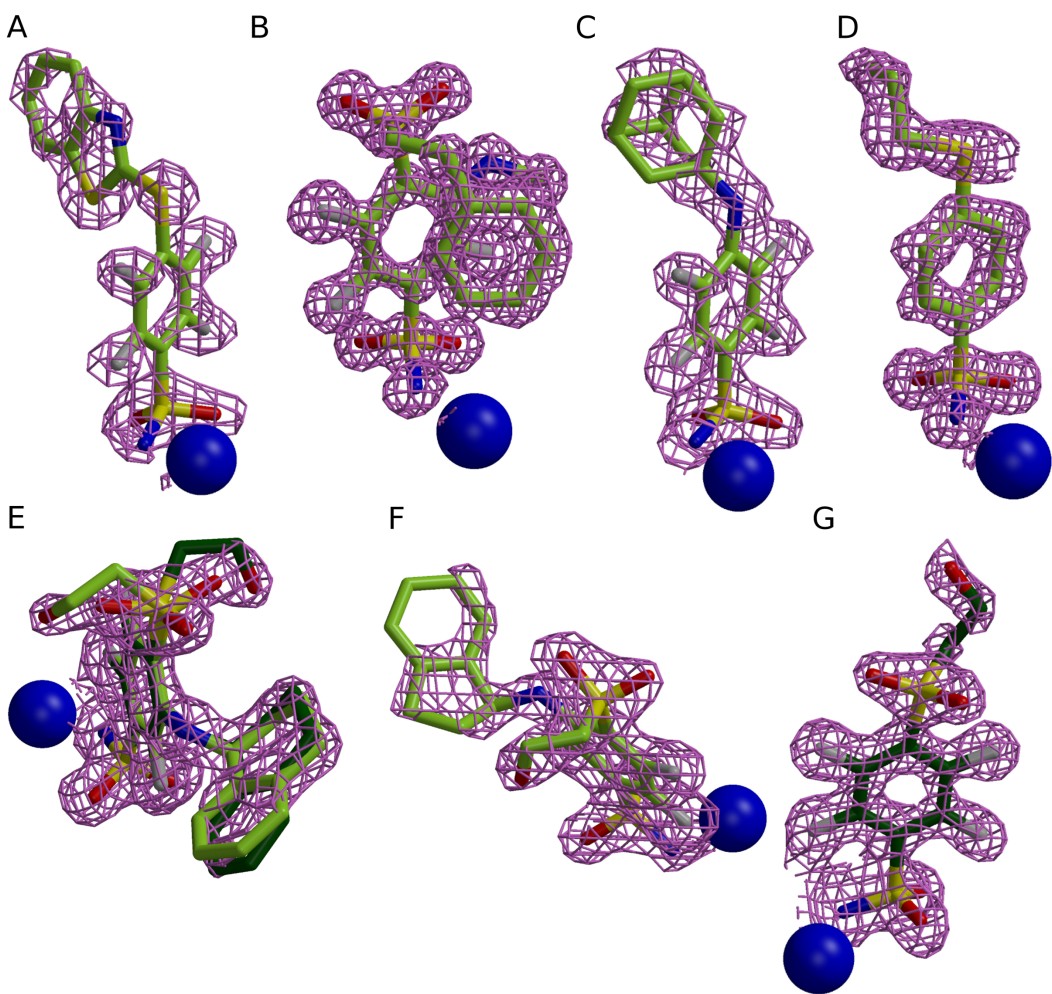

**Figure 1  Electron densities of seven compounds described in this study bound to different CA isoforms.** The electron densities of the compounds bound to CA II, CA XII, and CA XIII isoforms are shown as meshed isosurface and $Zn^{2+}$—as blue spheres. The electron density maps $|F_{obs} - F_{calc}|$ have been calculated after ligand removal from the crystal structure of CA-inhibitor complex are contoured to $3\sigma$, except (C) (contoured to $2.5\sigma$). (A) Compound **4** in CA II (PDB ID 5LLH). (B) Compound **13** in CA II (PDB ID 5LLC). (C) Compound **2** CA II (PDB ID 5LLE). (D) Compound **1** in CA II (PDB ID 5LLG). (E) Compound **16** in CA XII (PDB ID 5LLP, A subunit). (F) Compound **12** in CA XII (PDB ID 5LLO, A subunit). (G) Compound **10** in CA XII (PDB ID 5MSB, A subunit).

**12**—0.17 Å, **13**—0.24 Å, **9**—0.25 Å, **2**—0.22 Å, **14**—0.21 Å, **6**—0.26 Å); for CA I complexes vs apoCA I—0.17 ± 0.03 Å (compound **3**—0.14 Å and **5**—0.20 Å); for CA XII complexes vs apoCA XII—0.27 ± 0.10 Å (compound **10**—0.26 Å, **11**—0.52 Å, **1**—0.18 Å, **15**—0.32 Å, **16**—0.18 Å and **12**—0.17 Å) and for CA XIII complexes vs apoCA XIII—0.51 ± 0.04 Å (compound **9**—0.54 Å, **10**—0.46 Å and **11**—0.53 Å). The differences between main chain atoms between different crystal structures of the same protein have been reported to be around 0.25–0.40 Å (*Chothia & Lesk, 1986*). RMSD values of peptide backbone between apo proteins and complexes with inhibitor were in range 0.17–0.51 Å and we conclude that no significant conformational changes occur upon ligand binding.

**Table 2  X-ray crystallographic data collection and refinement statistics.** All datasets were collected at 100 K, test set size was 10%.

| Isoform-ligand | CA II-4 (VD10-49) | CA II-13 (VD11-26) | CA II-2 (VD10-51) | CA II-1 (VD12-10) | CA XII-10 (VD10-35) | CA XII-12 (VD12-34) | CA XII-16 (VD12-23) |
|---|---|---|---|---|---|---|---|
| PDB ID | 5LLH | 5LLC | 5LLE | 5LLG | 5MSB | 5LLO | 5LLP |
| Data-collection statistics: | | | | | | | |
| Space group | $P12_11$ | $P12_11$ | $P12_11$ | $P12_11$ | $P12_11$ | $P12_11$ | $P12_11$ |
| Unit-cell parameters (Å) | $a = 42.2$, $b = 41.1$, $c = 71.9$, $\beta = 104.4$ | $a = 42.2$, $b = 41.1$, $c = 72.0$, $\beta = 104.1$ | $a = 42.1$, $b = 41.0$, $c = 71.6$, $\beta = 104.0$ | $a = 42.1$, $b = 41.1$, $c = 71.8$, $\beta = 104.2$ | $a = 77.1$, $b = 74.1$, $c = 91.6$, $\beta = 108.9$ | $a = 77.3$, $b = 74.1$, $c = 91.6$, $\beta = 109.0$ | $a = 77.5$, $b = 74.3$, $c = 91.7$, $\beta = 109.0$ |
| Resolution (Å) | 1.9–14.8 | 1.1–40.9 | 1.9–14.2 | 1.1–39.7 | 1.3–74.0 | 1.6–73.1 | 1.5–73.3 |
| Wavelength (Å) | 1.5418 | 0.826606 | 1.5418 | 0.975522 | 0.976300 | 0.826606 | 0.976300 |
| Radiation source | Rigaku MicroMax ™-007 HF | EMBL, P13 | Rigaku MicroMax ™-007 HF | EMBL, P14 | EMBL, P14 | EMBL, P13 | EMBL, P14 |
| Unique reflections number | 17,591 | 95,940 | 18,840 | 83,155 | 237,650 | 127,546 | 160,218 |
| $R_{merge}$, overall (outer shell) | 0.160 (0.211) | 0.038 (0.265) | 0.109 (0.390) | 0.068 (0.400) | 0.056 (0.398) | 0.049 (0.370) | 0.062 (0.256) |
| I/ $\sigma$ overall (outer shell) | 7.0 (3.3) | 18.4 (5.7) | 17.6 (7.6) | 11.2 (3.3) | 14.2 (4.2) | 18.1 (4.0) | 15.7 (7.3) |
| Multiplicity overall (outer shell) | 2.1 (2.1) | 6.7 (6.5) | 3.0 (2.9) | 6.6 (5.5) | 6.8 (6.7) | 6.8 (6.9) | 6.9 (6.9) |
| Completeness (%) overall (outer shell) | 92.6 (92.6) | 98.8 (97.9) | 99.6 (100.0) | 91.5 (67.0) | 99.4 (99.2) | 99.0 (99.1) | 98.1 (96.6) |
| Wilson B-factor | 16.0 | 10.0 | 15.7 | 10.7 | 14.1 | 18.5 | 12.3 |
| Refinement statistics: | | | | | | | |
| $R_{work}$ | 0.182 | 0.138 | 0.159 | 0.138 | 0.149 | 0.185 | 0.164 |
| $R_{free}$ | 0.242 | 0.161 | 0.215 | 0.167 | 0.197 | 0.225 | 0.199 |
| RMSD bond lengths (Å) | 0.019 | 0.024 | 0.018 | 0.024 | 0.021 | 0.022 | 0.024 |
| RMSD bond angles (°) | 1.976 | 2.408 | 1.891 | 2.307 | 2.183 | 2.146 | 2.339 |
| Average B factors (Å²): | | | | | | | |
| All | 15.82 | 17.65 | 15.44 | 17.79 | 21.56 | 19.31 | 15.62 |
| Main-chain | 14.21 | 13.59 | 13.17 | 14.13 | 17.93 | 16.61 | 12.21 |
| Side-chain | 16.03 | 18.14 | 15.54 | 18.79 | 22.89 | 19.83 | 16.05 |
| Inhibitors | 15.45 | 12.90 | 27.07 | 21.42 | 22.45 | 23.34 | 22.53 |
| Waters | 21.24 | 30.20 | 23.28 | 29.41 | 30.28 | 27.60 | 25.57 |
| Zinc | 7.14 | 7.26 | 6.93 | 8.15 | 10.44 | 10.98 | 6.09 |
| Other molecules | 43.05 | 30.64 | 42.42 | 35.04 | 18.08 | 29.47 | – |
| Number of atoms: | | | | | | | |
| All | 2,265 | 2,562 | 2,297 | 2,520 | 9,823 | 9,755 | 9,957 |
| Protein | 2,058 | 2,209 | 2,065 | 2,166 | 8,606 | 8,570 | 8,619 |
| Inhibitor | 24 | 26 | 25 | 28 | 100 | 116 | 150 |
| Water | 156 | 302 | 184 | 306 | 1,082 | 1,053 | 1,184 |
| Zinc | 1 | 1 | 1 | 1 | 4 | 4 | 4 |
| Other molecules | 26 | 24 | 22 | 19 | 31 | 12 | 0 |
| Ramachandran statistics (%): | | | | | | | |
| Most favored regions | 95 | 96 | 96 | 97 | 97 | 98 | 98 |
| Additionally allowed regions | 5 | 4 | 4 | 3 | 3 | 2 | 2 |
| Outliers | 0 | 0 | 0 | 0 | 0 | 0 | 0 |

## Calculation of molecular surface

The buried surface area (BSA) and accessible surface area (ASA) values were calculated using Voronota (v. 1.10.1544, *Olechnovič & Venclovas, 2014*) program. Hydrogen atoms were added to models by PERL script. The initial parameters for contacts calculation were set as follows: (1) rolling probe radius was default (1.4 Å); (2) values of Van-der-Waals radii were the following: Cl atom—1.75 Å, C—1.7 Å, F—1.47 Å, N—1.55 Å, O—1.52 Å, S—1.80 Å, Zn—1.39 Å, and H—1.09 Å.

## Isothermal titration calorimetry

ITC experiments for the binding of **13–14** and **16** to CA II, CA XII and CA XIII were performed using VP-ITC instrument (Microcal, Inc., Northampton, MA, USA, now part of Malvern Instruments Ltd, UK) with 5–10 μM protein solution in the cell and 50–100 μM compound solution in the syringe. A typical experiment consisted of 25 injections (10 μl each) within 3–4 min intervals. Experiments were carried out at 37 °C in a 50 mM sodium phosphate or TRIS chloride buffer containing 100 mM NaCl at pH 7.0, with a final DMSO concentration of 1%, equal in the cell and syringe. Protein stock solutions were dialyzed against the buffers that were used to prepare the ligand solutions. ITC data were analyzed using MicroCal Origin software. All calorimetric titration data were presented after subtracting the background signal. The first point from the 5 μL injection in the integrated data graph was deleted. The binding constants, enthalpies and entropies of binding were estimated after fitting the data with the single binding site model.

## Determination of intrinsic thermodynamics of binding

The *observed* energies of binding, including the Gibbs energies (*observed* affinities), enthalpies, and entropies of the protein-ligand pairs were determined by the fluorescent thermal shift assay (*Pantoliano et al., 2001*; *Lo et al., 2004*; *Matulis et al., 2005*; *Cimmperman & Matulis, 2011*) and isothermal titration calorimetry (*Wiseman et al., 1989*; *Harding & Chowdhry, 2001*; *Broecker, Vargas & Keller, 2011*). The measured *observed* thermodynamic parameters of binding depend on the assay conditions such as pH and buffer because the binding of sulfonamide to CA is linked to several protonation reactions, the protonation of hydroxide bound to the Zn(II) in the active site of CA, the deprotonation of sulfonamide group of the inhibitor, and the (de)protonation of the buffer (*Baker & Murphy, 1996*; *Krishnamurthy et al., 2008*; *Baranauskienė & Matulis, 2012*; *Morkūnaitė et al., 2015*). For the structure–activity relationship and correlation of thermodynamic parameters with the X-ray structures, the *intrinsic* thermodynamic parameters of sulfonamide anion binding to the Zn-bound water form of CA were calculated.

The intrinsic binding thermodynamics of compounds **1–12**, and **15** were taken from references given in the Table 3. Isothermal titration calorimetry (ITC) was used to determine the enthalpy of **13–14** and **16** binding to CAII, CA XII, and CA XIII (the intrinsic affinities, determined by the fluorescent thermal shift assay, were taken from *Dudutienė et al. (2015)*. We have shown previously (*Smirnovienė, Smirnovas & Matulis, 2017*) that the $K_b$ values observed by FTSA and ITC methods were in good agreement and varied up to 3 times. However, the $K_b$ determined by ITC method were not used in this study, because some

**Table 3   The CA isoform—inhibitor binding intrinsic thermodynamic parameters.** PDB IDs and corresponding references are listed for all CA-compound complexes where the crystallographic structures were solved. The intrinsic binding data were taken from publications *Kišonaitė et al. (2014)*, *Zubrienė et al. (2015)* and *Zubrienė et al. (2016)*.

| CA and inhibitor | PDB ID | $K_d$, nM, FTSA | $\Delta G$, kJ/mol | $\Delta H$, kJ/mol | $-T\Delta S$, kJ/mol | References of crystal structures |
|---|---|---|---|---|---|---|
| CA II-**1** | 5LLG | 0.0074 | −66.1 | −69.0 | 2.9 | 7 |
| CA XII-**1** | 4WW8 | 0.067 | −60.4 | −56.4 | −4.0 | 1 |
| CA II-**2** | 5LLE | 0.060 | −60.7 | −25.1 | −35.6 | 7 |
| CA I-**3** | 4WR7 | 0.0021 | −69.4 | −74.4 | 5.0 | 1 |
| CA II-**3** | 4WW6 | 0.062 | −60.6 | −46.4 | −14.2 | 1 |
| CA XII-**3** | 5MSA[a] | 1.14 | −53.1 | −31.7 | −21.4 | – |
| CA XIII-**3** | 5LLN[a] | 0.099 | −59.4 | −54.3 | −5.1 | – |
| CA II-**4** | 5LLH | 0.067 | −60.4 | −37.7 | −22.8 | 7 |
| CA I-**5** | 4WUQ | 0.0015 | −70.3 | −59.7 | −10.6 | 1 |
| CA II-**6** | 3SBH | 0.080 | −59.9 | −48.4 | −11.5 | 2 |
| CA II-**7** | 3SBI | 0.025 | −63.0 | −62.0 | −1.0 | 2 |
| CA II-**8** | 4KNI | 0.085 | −59.8 | −41.1 | −18.7 | 3 |
| CA XIII-**8** | 4KNM | 0.18 | −57.9 | −51.6 | −6.3 | 3 |
| CA II-**9** | 4KNJ | 0.11 | −59 | −47.6 | −11.4 | 3 |
| CA XII-**9** | 4KP5[a] | 3.0 | −50.6 | −32 | −18.6 | 3 |
| CA XIII-**9** | 4KNN | 0.31 | −56.5 | −54.8 | −1.7 | 3 |
| CA II-**10** | 4PZH | 2.7 | −50.9 | −46.7 | −4.2 | 5 |
| CA XII-**10** | 5MSB | 35 | −44.3 | −24.6 | −19.7 | 7 |
| CA XIII-**10** | 4HU1 | 8.9 | −47.8 | −50.4 | 2.7 | 8 |
| CA I-**11** | 5E2M[a] | 60 | −42.9 | 0.8 | −43.7 | 6 |
| CA II-**11** | 4PYY[a] | 2.4 | −51.2 | −9.5 | −41.7 | 5 |
| chimeric CA IX-**11** | 4Q07[a] | n/d | n/d | n/d | n/d | 5 |
| CA XII-**11** | 4Q0L | 0.11 | −59.0 | −40.8 | −18.2 | 5 |
| CA XIII-**11** | 5E2N | 0.29 | −56.6 | −11.4 | −45.2 | 6 |
| CA II-**12** | 5DRS | 1.1 | −53.2 | −36.4 | −16.8 | 6 |
| CA XII-**12** | 5LLO | 0.4 | −55.8 | −47.0 | −8.8 | 7 |
| CA II-**13** | 5LLC | 0.3 | −56.5 | −49.4 | −7.1 | 7 |
| CA II-**14** | 4QJM | 0.34 | −56.2 | −23.1 | −33.1 | 4 |
| CA XII-**14** | 4QJO[a] | n/d | n/d | n/d | n/d | 4 |
| CA XIII-**14** | 4QJP[a] | 0.13 | −56.9 | −17.7 | −39.1 | 4 |
| CA II-**15** | 4QTL[a], 5EHE[a] | 4.5 | −49.5 | −22.1 | −27.4 | 4, 6 |
| CA XII-**15** | 4QJW | 1.2 | −53.0 | −36.1 | −16.9 | 4 |
| CA XIII-**15** | 4QJX[a] | 1.6 | −52.2 | −34.2 | −18.0 | 4 |
| CA XII −**16** | 5LLP | 0.58 | −54.8 | −29.8 | −25 | 7 |

**Notes.**
1-*Zubrienė et al. (2015)*; 2-*Čapkauskaitė et al., (2012)*; 3-*Čapkauskaitė et al. (2013)*; 4-*Dudutienė et al. (2015)*; 5-*Dudutienė et al. (2014)*; 6-*Zubrienė et al. (2016)*; 7- This study; 8- *Davis et al. (2013)*.
[a] PDB ID entries of the crystal structures that are not analyzed in this article.

compounds bound CAs with observed $K_d < 10$ nM and Wiseman c factor was too large for accurate $K_b$ determination. To avoid discrepancies only the $K_b$ determined by FTSA were used.

The intrinsic thermodynamic parameters for compounds **13–14** and **16** were calculated using Eqs. (1)–(4), where the p$K_a$ of sulfonamide protonation was 7.88 and the enthalpy of sulfonamide deprotonation $-24.3$ kJ/mol.

The intrinsic binding constant $K_{b\_intr}$ is equal to the observed binding constant $K_{b\_obs}$ divided by the fractions of deprotonated inhibitor and protonated Zn-bound water form of CA:

$$K_{b\_int} = \frac{K_{b\_obs}}{f_{RSO_2NH^-} * f_{CAZnH_2O}}. \tag{1}$$

The fractions of the deprotonated inhibitor and the Zn bound water form of CA may be calculated:

$$f_{RSO_2NH^-} = \frac{10^{pH-pK_{a\_sulf}}}{1+10^{pH-pK_{a\_sulf}}} \tag{2}$$

$$f_{CAZnH_2O} = 1 - \frac{10^{pH-pK_{a\_ZnH_2O}}}{1+10^{pH-pK_{a\_ZnH_2O}}}. \tag{3}$$

The observed enthalpy ($\Delta_b H_{obs}$) is the sum of all protonation events and the intrinsic binding reaction:

$$\Delta_b H_{intr} = \Delta_b H_{obs} - n_{sulf}\Delta_{b\_proton\_sulf}H - n_{CA}\Delta_{b\_proton\_CA}H + n_{buf}\Delta_{b\_proton\_buf}H \tag{4}$$

where $\Delta_b H_{obs}$ is the observed binding enthalpy, $n_{sulf} = f_{RSO_2NH^-} - 1$ is the number of protons uptaken by the inhibitor, $\Delta_{b\_proton\_sulf}H$ is the enthalpy of inhibitor protonation, $n_{CA} = 1 - f_{CAZnH_2O}$ is the number of protons uptaken by the Zn-hydroxide, $\Delta_{b\_proton\_CA}H$ is the enthalpy of CA protonation, $n_{buf} = n_{sulf} + n_{CA}$, is the net sum of uptaken or released protons and $\Delta_{b\_proton\_buf}H$ is the enthalpy of buffer protonation.

## RESULTS AND DISCUSSION

A series of 24 crystal structures of complexes between four CA isoforms and 16 aromatic benzenesulfonamides were solved by X-ray crystallography characterizing the structural aspect of the interaction. To characterize the same interactions energetically, the *observed* affinities of binding (*observed* dissociation constants, $K_d$, or *observed* Gibbs energies of binding) for each protein-ligand pair were determined by the fluorescent thermal shift assay (FTSA), and the *observed* enthalpies were determined by isothermal titration calorimetry (ITC). The $K_d$ by both methods (FTSA and displacement ITC) agreed nearly perfectly (*Zubrienė et al., 2015*). The *observed* entropies (multiplied by the absolute temperature) were obtained by subtracting the Gibbs energies obtained by FTSA from the enthalpies obtained by ITC. However, sulfonamide ligand binding to CA is linked to several protonation reactions and therefore the *observed* values depend on pH and buffer. To dissect these linked reactions, the *intrinsic* thermodynamic parameters that are independent of the buffer and pH of deprotonated sulfonamide anion binding to the Zn-bound water form

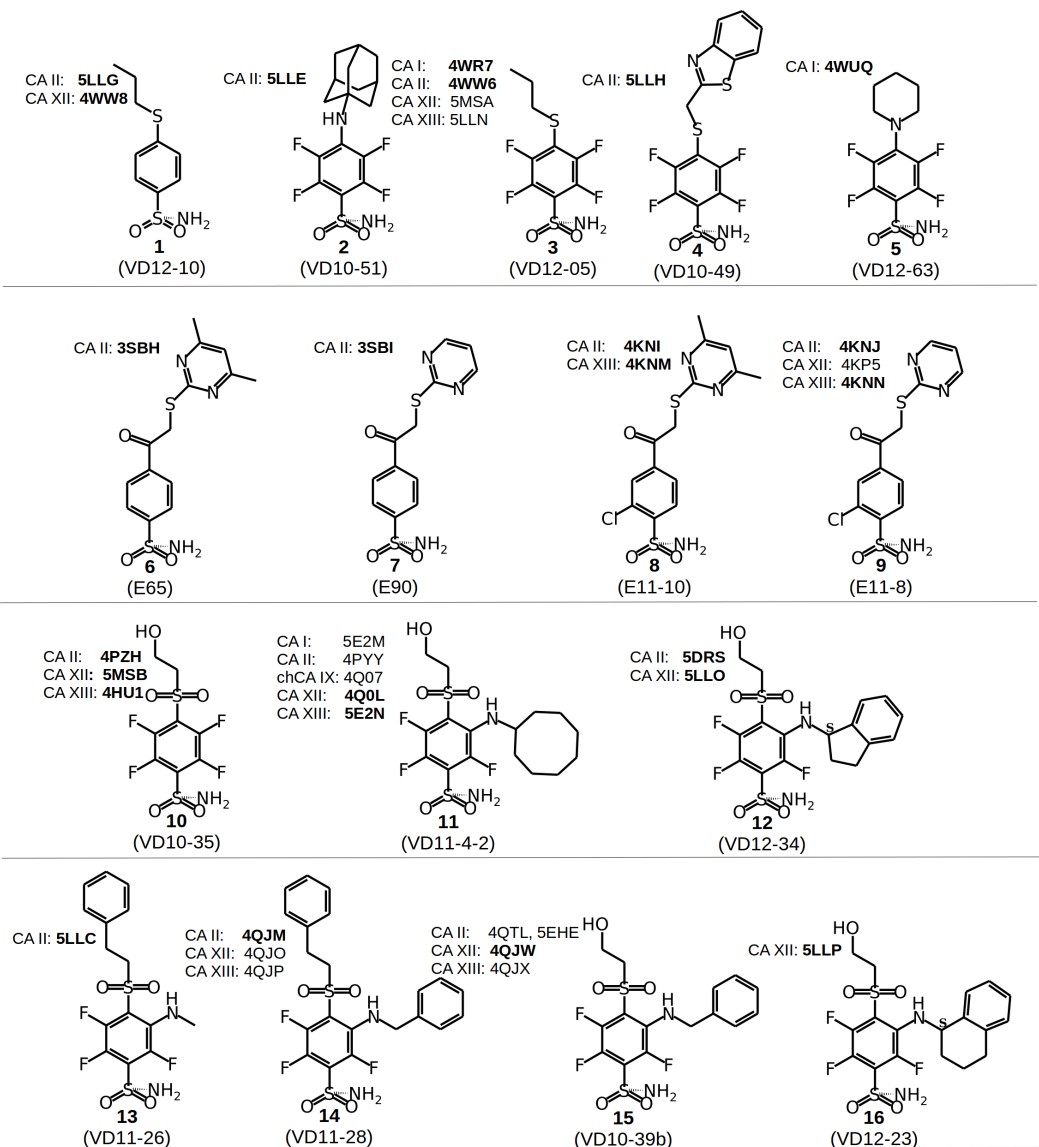

**Figure 2 Chemical structures of CA inhibitors in the crystal structures of CA isoforms described in this study.** PDB IDs are given in bold. Compound numbers and conventional names are listed below each structure.

of CA were calculated (see Methods). This analysis of structure–activity relationship and correlations of *intrinsic* thermodynamic parameters with the X-ray structures characterized each protein-ligand pair both structurally and thermodynamically.

Chemical structures of the 16 benzenesulfonamides used in this study are shown in Fig. 2. Crystal structures of these compounds with CA I, CA II, CA XII, and CA XIII are grouped into pairs combining chemically similar compounds that differ by the hydrophobicity of substituents whose contribution to the binding to CA isoforms is being studied. Fifteen inhibitors were grouped into 13 matched pairs of crystal structures. Six pairs contain crystal

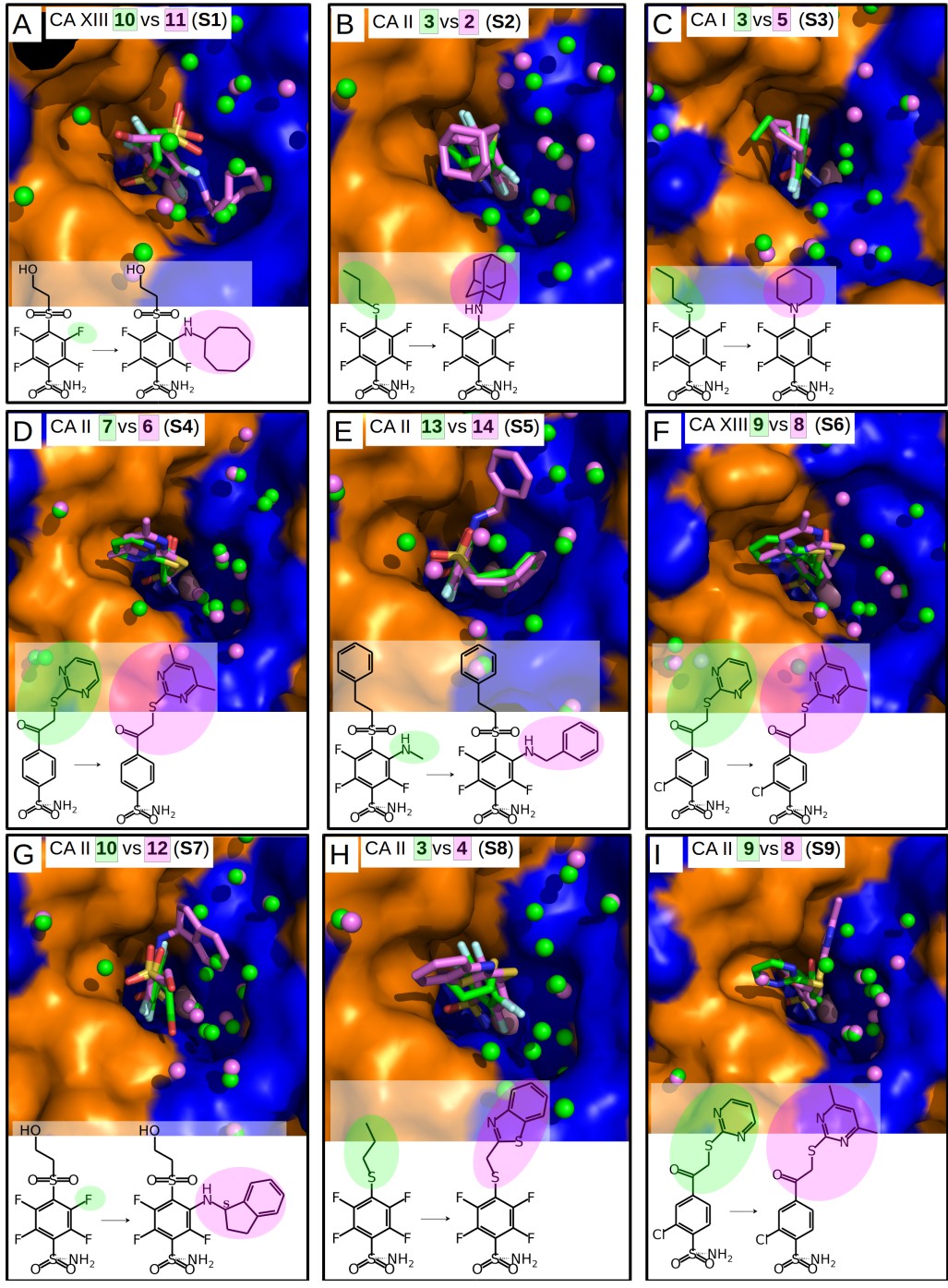

**Figure 3** **"Similar binders": matched pairs of inhibitors bound in the crystal structures of CA iso-forms.** Inhibitors bound in CA active sites are shown in same orientation. The structural differences of the inhibitors in pairs are shaded by green or pink. The ligands in crystal structures are colored accordingly. Water molecules found in crystal structures are shown as green and pink spheres. $Zn^{2+}$ ion is shown as a magenta sphere. (continued on next page...)

**Figure 3 (...continued)**
The protein surface of CA active site is colored orange for hydrophobic residues (Val, Ile, Leu, Phe, Met, Ala, Gly, and Pro) and blue for the residues with charged and polar side chains (Arg, Asp, Asn, Glu, Gln, His, Lys, Ser, Thr, Tyr, Trp, and Cys). (A) Compounds 10 (green, PDB ID 4HU1) and 11 (pink, PDB ID 5E2N) bound in CA XIII (pair S1). (B) Compounds 3 (green, PDB ID 4WW6) and 2 (pink, PDB ID 5LLE) bound to CA II (pair S2). (C) Compounds 3 (green, PDB ID 4WR7) and 5 (pink, PDB ID 4WUQ) bound to CA I (pair S3). (D) Compounds 7 (green, PDB ID 3SBI) and 6 (pink, PDB ID 3SBH) bound to CA II (pair S4). (E) Compounds 13 (green, PDB ID 5LLC) and 14 (pink, PDB ID 4QJM) bound to CA II (pair S5). (F) Compounds 9 (green, PDB ID 4KNN) and 8 (pink, PDB ID 4KNM) bound to CA XIII (pair S6). (G) Compounds 10 (green, PDB ID 4PZH) and 12 (pink, PDB ID 5DRS) bound to CA II (pair S7). (H) Compounds 3 (green, PDB ID 4WW6 ) and 4 (pink, PDB ID 5LLH ) bound to CA II (pair S8). (I) Compounds 9 (green, PDB ID 4KNJ) and 8 (pink, PDB ID 4KNI) bound to CA II (pair S9).

structures with CA II, two—with CA XIII, one—with CA I, and four—with CA XII. Two crystal structures of the compound 1 were compared when bound in the active site of CA II and CA XII.

Nine matched pairs of crystal structures exhibited a *similar* binding mode (pairs S1-9) of the inhibitors in the active site of CA isoforms, i.e., the benzene ring was found in the same orientation while only the positions of substituents may have differed in each pair (Fig. 3). The remaining four matched pairs of complexes exhibited a *dissimilar* binding mode of the benzenesulfonamide ring in the active site of CA (D1-4) (Figs. 4A–4D).

### *Similar* binders

The sulfonamide-bearing benzene ring was observed in the same orientation for all pairs of "*similar* binders" (S1-9, Fig. 3). Differences in the binding of ligands within these pairs were observed only in the regions where different substituents were present in the compounds of the pair. The *intrinsic* thermodynamic parameters of binding are listed in Table 3. Figure 5A shows the energies, including the Gibbs energy, enthalpy, and entropy of binding, for every analyzed pair of similar binders, as a function of the sum of accessible and buried surface areas of the hydrophobic substituents of inhibitors in each matched pair. The accessible and buried surface areas of substituent groups, marked green or magenta in Fig. 3, were calculated by VORONOTA algorithm (Olechnovič & Venclovas, 2014).

The comparison of intrinsic Gibbs energy of binding for each molecular pair in the group S1-9 (Fig. 5A) showed that there was a relatively small increase in affinity with an increase in the accessible and buried surface area. Additional hydrophobic surface seemed to not have a large impact on the binding affinities. However, the intrinsic enthalpies of binding in all pairs of similar binders were significantly less exothermic for compounds bearing larger hydrophobic substituents. Additional contacts between the protein and the ligand did not make the enthalpy more favorable. The entropies of binding mirrored the enthalpies—the enthalpy-entropy compensation was observed (Fig. 5A).

The compounds 2, 3, and 4 are *para*-substituted fluorinated benzenesulfonamides exhibiting nearly identical binding affinity for CA II (Figs. 3B and 3H, pairs S2 and S8). Fluorinated benzene rings of these compounds bind to the active site of CA II in almost identical mode and their hydrophobic *para*-substituents interact with the hydrophobic cavity formed by the amino acids Phe131, Val135, Leu198, Pro202, and Leu204. The properties of *para*-substituents in these compounds are different: propyl of compound 3

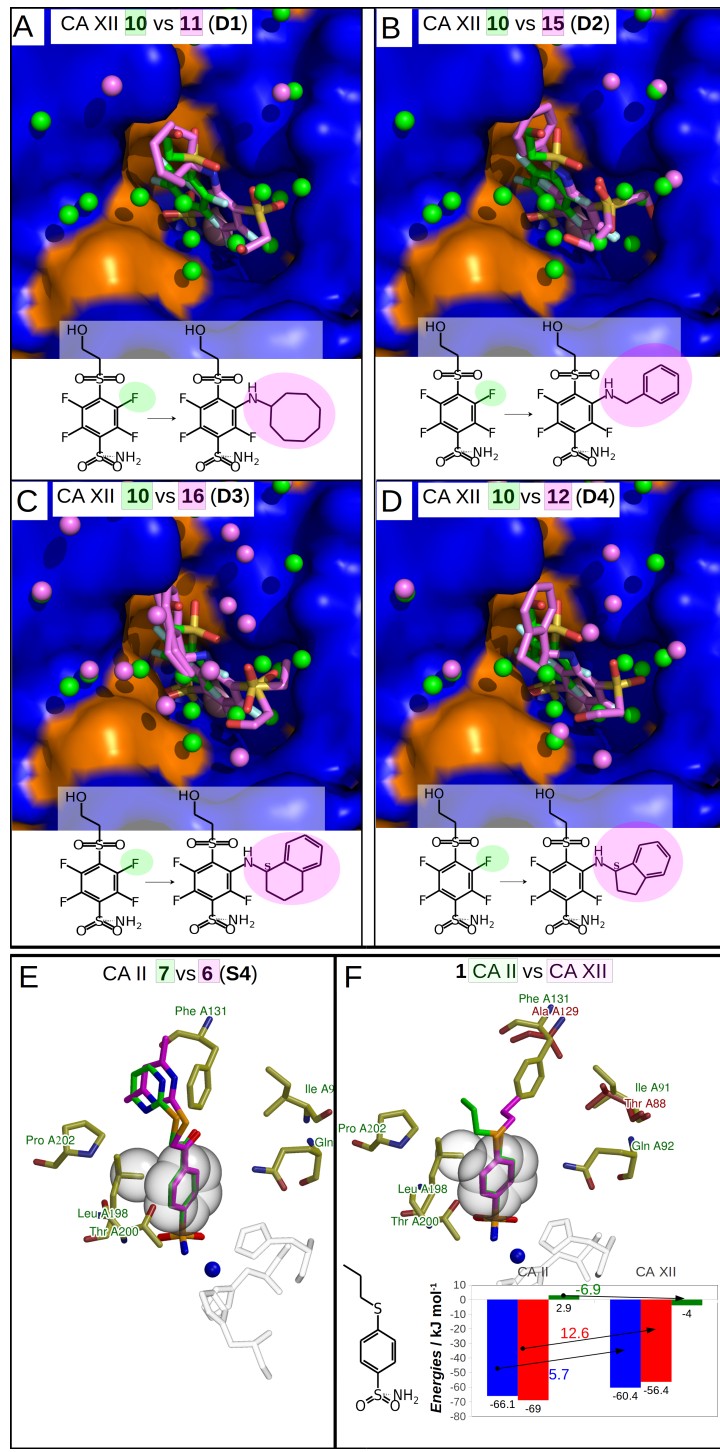

**Figure 4** **Dissimilar binders.** (A–D) "Dissimilar binders": matched pairs of crystal structures. Inhibitors bound to CAs are presented in the same orientation. The structural differences of the inhibitors in pairs are shaded by green or pink. The ligands in crystal structures are (continued on next page…)

**Figure 4 (…continued)**
colored accordingly. Water molecules found in crystal structures are shown as green and pink spheres. $Zn^{2+}$ ion is shown as a magenta sphere. The protein surface of CA active site is colored orange for hydrophobic residues (Val, Ile, Leu, Phe, Met, Ala, Gly, and Pro) and blue for the residues with charged and polar side chains (Arg, Asp, Asn, Glu, Gln, His, Lys, Ser, Thr, Tyr, Trp, and Cys). (A) Compounds **10** (green, PDB ID 5MSB) and **11** (pink, PDB ID 4Q0L) are bound to CA XII (pair **D1**). (B) Compounds **10** (green, PDB ID 5MSB) and **15** (pink, PDB ID 4QJW) bound to CA XII (pair **D2**). (C) Compounds **10** (green, PDB ID 5MSB) and **16** (pink, PDB ID 5LLP) bound to CA XII (pair **D3**). (D) Compounds **10** (green, PDB ID 5MSB) and **12** (pink, PDB ID 5LLO) bound to CA XII (pair **D4**). (E) Compounds **7** (green, PDB ID 3SBI) and **6** (pink, PDB ID 3SBH) bound to CA II (colored yellow). (F) Comparison of the compound **1** position bound to CA II (**1** is green, CA II is colored yellow, PDB ID 5LLG) and CA XII (**1** is pink, CA XII side chains are red, PDB ID 4WW8) and corresponding binding thermodynamics. The histogram in the insert shows the comparison of binding thermodynamics: $\Delta G$ is blue, $\Delta H$—red and $-T\Delta S$—green. The histidine side chains holding the active site $Zn^{2+}$ (blue sphere) ion are transparent in (E) and (F). The Leu198 conservative between isoforms CA II and XII and benzene rings of compounds are shown in CPK representation which marked the specific aliphatic-aromatic interaction.

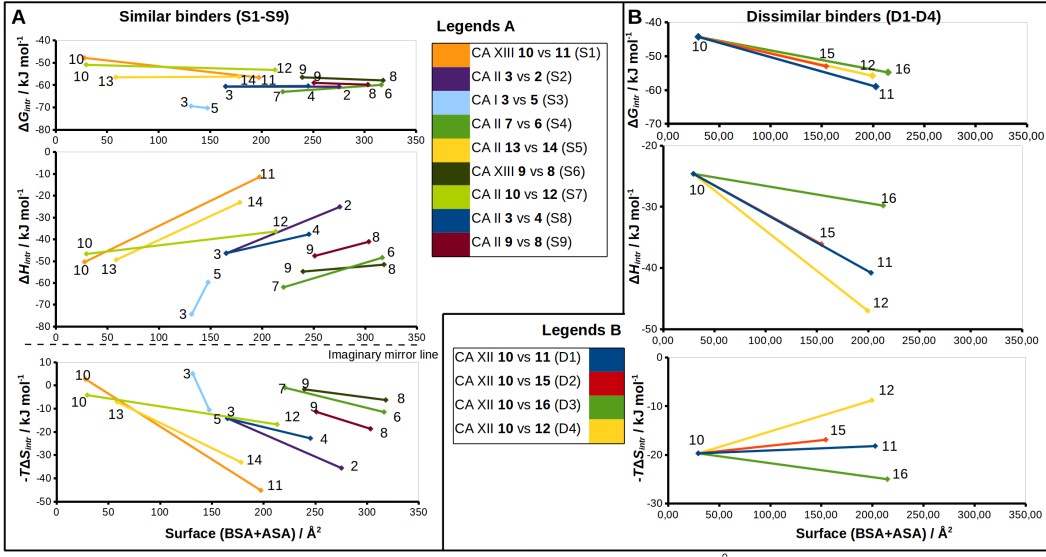

**Figure 5** Correlations between structural changes and the intrinsic thermodynamics of binding to CA isoforms in the matched pairs for *similar* (**A**) and *dissimilar* binders (**B**). The surface areas (BSA + ASA, in $Å^2$) of the functional groups marked by colored circles (Figs. 2 and 3) are shown on *x*-axis. The values of the surface area is the sum of 'voronota' contacts (*Olechnovič & Venclovas, 2014*) (buried accessible surface) and the solvent accessible surface. The thermodynamic parameters are shown on *y*-axis. Colored lines connect two compounds in the pair with the numbers of ligands at the ends of each line. Due to the enthalpy-entropy compensation effect described for similar binders (Figs. 3A and 3I, pairs **S1–S9**), the intrinsic entropies mirrored the enthalpies as emphasized by the imaginary mirror plane (dashed line) for the similar binders in (A).

is small and flexible and can adopt an optimal conformation for interactions, while **2** and **4** have large and rigid adamantane and aromatic heterocyclic groups, respectively. The hydrophobic area of the *para*-substituents increases in **4** and **2** as compared to **3** and the entropic gain of the corresponding binding reaction increases too (Fig. 5A and Table 3). The identical binding mode of these compounds allows to propose that $\Delta H_{interactions}$ for

these compounds can be similar in the binding to CA II. $\Delta H_{\text{conformational}}$ gain also can be similar due to (1) identical binding mode to CA II and (2) congeniality between these compounds. As mentioned previously, the pH and buffer influence to $\Delta H_{\text{exchange}}$ gain is eliminated because we analyze the intrinsic binding parameters. We hypothesize that the enthalpy-entropy differences are caused by the reorganization of the solvent near the hydrophobic surfaces of the compound, but not of compounds themselves. The changes of $\Delta H_{\text{desolvation}}$ and $-T\Delta S_{\text{desolvation}}$ seem as first candidates for explaining the changes between binding thermodynamics of these compounds. The changes of desolvation entropy can be explained by increase of translational and rotational mobility (freedom) of water molecules which were more ordered near the hydrophobic surfaces of compounds before desolvation.

The active site of crystal structure CA II-**3** contains more water molecules in comparison with CA II-**2** and CA II-**4** (Figs. 3**B** and 3**H**). However, it is difficult to make accurate analysis of the water molecules because the number of solvent molecules that are visible in crystal structure depend on the resolution. There are more water molecules in the crystal structure CA II–**3** with the resolution of 1.06 Å (17 waters by Pymol searching around the ligand within 9 Å distance) as compared to CA II-**4** (1.90 Å resolution, six waters) and CA II- **2** (1.90 Å resolution, nine waters). Furthermore, the active site of crystal structure CA II-**3** possibly contains more water molecules than it is currently observed in comparison with structures CA II-**2** and CA II-**4** due to the only one distinct conformation of the flexible His64 (which is important for the rate limiting proton transfer event from the active site). The active sites of crystal structures CA II-**2** and CA II-**4** contain electron densities of two conformations of His64, one of which disturbs the quality of the electron densities of water molecules in the hydrophilic part of the active site.

Compounds **3** vs **5** in the matched pair **S3** are *para*-substituted fluorinated benzenesulfonamides bound in identical position in the CA I active site (Fig. 3**C**, pair **S3**). The binding of **3** by CA I described in *Dudutienè et al. (2015)* with an unfavorable entropic contribution (Table 3). The extremely high binding affinity was explained by the specific interactions between **3** and CA I where the fluorobenzene ring is fixed between His200 and Leu198 which restricts the mobility of fluorobenzene ring of the ligand from the other side by the aromatic-aliphatic interaction. The minor enlargement of the hydrophobic area (cyclohexane in **5** vs. propyl in **3**) resulted in significant changes in the enthalpy-entropy contribution ($\Delta\Delta G = -0.9 \pm 1$, $\Delta\Delta H = 14.7$, $-\Delta T\Delta S = -15.6$ kJ/mol). The same binding affinities in the pair indicate that compounds interact with the active site in a similar way (similar $\Delta H_{\text{interactions}}$) and the enlargement of the hydrophobic surface is the only factor that influences the difference in thermodynamics.

The unfavorable entropic contribution of **3** binding to CA I seem as uncompensated unfavorable changes of conformational entropy ($-T\Delta S_{\text{conformational}}$) due to the reduction in mobility of binding partners upon formation of complex.

Compounds **13** and **14** are *meta/para*-substituted fluorinated benzenesulfonamides that differ by *meta*-substituents. Both compounds have *para*-phenyl groups that were found in both crystal structures with CA II "stacked" with the fluorinated benzene ring. In such conformation, the *para*-phenyl group in both ligands dislocates the water molecules

from the hydrophilic part of the active site of CA II (Fig. 3E, S5). The binding of **13** with CA II is enthalpy-driven and the release of water molecules from the active site did not make the entropy more favorable (Table 3, Fig. 5A) due to increase in translational and rotational mobility of waters released from the active site. The additional hydrophobic area of *meta*-group for the compound **14** induces large changes in enthalpy-entropy terms ($\Delta\Delta G = 0.3 \pm 1$, $\Delta\Delta H = 26.3$, $-\Delta T\Delta S = -26$ kJ/mol, Table 3). On the other hand, the overall binding affinity is the same for both compounds.

The structural differences of **10** vs **12** bound in CA II (Fig. 3G, S7) are located in the *meta*-position of the benzene ring. These compounds have the same position of the fluorinated benzenesulfonamide and of sulfonyl moiety of *para*-group. The enlargement of the hydrophobic part induces the increase of the binding affinity by $-2.3 \pm 1$ kJ/mol (Table 3) within this pair. The changes of the entropic and enthalpic contributions of the pair **S7** ($\Delta\Delta G = -2.3 \pm 1$, $\Delta\Delta H = 10.3$, $-\Delta T\Delta S = -12.6$ kJ/mol) are more than two times smaller than in the pair **S5** (**13** and **14** in CA II, Fig. 5A), despite somewhat larger variations in the surface between **10** and **12**. The enthalpy-entropy compensation takes place in both cases.

The pair of structures **S1**, where compounds **10** and **11** (Fig. 3A), that differ by the presence of the bulky cyclooctyl group in the *meta*-position (Fig. 2) are bound to CA XIII, is an interesting exception between reviewed pairs: the binding affinity is notably increased with the increase of the hydrophobic surface of the *meta*-group (Fig. 5A). Inhibitor **11** binds to CA XIII 30 times stronger than **10** ($K_d$ 0.29 nM for **11** and 8.9 nM for **10**, Table 3). The cyclooctyl group was found in the crystal structure bound to the hydrophilic part of the CA XIII active site displacing water molecules observed in the structure with compound **10** (Fig. 3A, S1). The entropic and enthalpic contributions are large for this pair ($\Delta\Delta G = -8.8 \pm 1$, $\Delta\Delta H = 39.0$, $-\Delta T\Delta S = -47.9$ kJ/mol). The compound **11** is entropy-driven ligand and large entropic gain seems to be obtained from the changes of desolvation entropy on the hydrophobic surface of ligand and maybe by the release of water molecules from the active site.

Compounds **9** and **8** are *para*-substituted 2-chlorobenzenesulfonamides, that contain pyrimidine and dimethylpyrimidine groups in the *para*-position, respectively (Fig. 2). We compared the crystal structures of this pair of inhibitors bound to CA II (Fig. 3I, pair **S9**) and CA XIII (Fig. 3F, pair **S6**). The increase of hydrophobic surface between **9** and **8** is small, it is an addition of two methyl groups to the *para*-pyrimidine. Therefore, for both pairs, the favorable impact to binding affinities is rather small and occurs due to negligible favorable changes of entropies vs changes of enthalpies (Table 3 and Fig. 5A). In CA II (pair **S9**): $\Delta\Delta G = -0.8 \pm 1$, $\Delta\Delta H = 6.5$, $-\Delta T\Delta S = -7.3$ kJ/mol and in CA XIII (pair **S6**): $\Delta\Delta G = -1.4 \pm 1$, $\Delta\Delta H = 3.2$, $-\Delta T\Delta S = -4.6$ kJ/mol. In CA XIII (pair **S6**), methylpyrimidine of compound **8** coincides with the pyrimidine ring of **9**, whereas in CA II (pair **S9**), methylpyrimidine of **8** interacts with the other hydrophobic part of the active site than pyrimidine ring of **9** (Figs. 3F and 3I). Methylpyrimidine is larger than pyrimidine and could not occupy the same position as pyrimidine in both CA isoforms. The more spacious hydrophobic part of the CA II active site allowed for the reorientation of the

methylpyrimidine along the hydrophobic surface whereas in CA XIII the methylpyrimidine is simply dislocated outwards.

Compounds **7** and **6** are homologous to compounds **9** and **8** respectively (Fig. 2), except that the benzene ring does not contain chlorine. This pair of ligands is represented by the crystal structures of the complexes with CA II (Fig. 3D, pair **S4**). The binding thermodynamics in **S4** exhibits the distinct decrease of the binding affinity towards CA II: $\Delta\Delta G = 3.1 \pm 1$, $\Delta\Delta H = 13.6$, and $-\Delta T\Delta S = -10.5$ kJ/mol due to the uncompensated enthalpic loss and the entropic gain (Fig. 5A and Table 3).

### *Dissimilar* binders

The thermodynamic parameters of the *dissimilar* inhibitor (Figs. 4A–4D) binding to CA XII are shown in Fig. 5B as a function of the surface of substituent at the benzenesulfonamide group. For the dissimilar binders, there is a significant increase of the intrinsic binding affinity between compounds in matched pairs (e.g., 318 times for the pair **D1**) when the accessible and buried surface areas increase. The intrinsic enthalpies of binding in all pairs of dissimilar binders were significantly more exothermic for compounds with larger hydrophobic substituents, contrary to the binding thermodynamics of similar binders (pairs **S1**–**S9**): pair **D1** ($\Delta\Delta G = -14.7 \pm 1$, $\Delta\Delta H = -16.2$, $-\Delta T\Delta S = 1.5$ kJ/mol), pair **D2** ($\Delta\Delta G = -8.7 \pm 1$, $\Delta\Delta H = -11.5$, $-\Delta T\Delta S = 2.8$ kJ/mol), pair **D3** ($\Delta\Delta G = -10.5 \pm 1$, $\Delta\Delta H = -5.2$, $-\Delta T\Delta S = -5.3$ kJ/mol), and pair **D4** ($\Delta\Delta G = -11.5 \pm 1$, $\Delta\Delta H = -22.4$, $-\Delta T\Delta S = 10.9$ kJ/mol). Additional contacts between the protein and the ligand made the enthalpy more favorable for this group. The entropy was significantly favorable for the pair **D3**, significantly unfavorable for **D4**, and the changes were negligible for **D1** and **D2**. The compounds that are more hydrophobic in these pairs (**15**, **16**, **12**, and **11**) differ by the *meta*-substituent from the reference compound **10** and have the same location of the fluorinated benzene ring in the CA XII active site. The *meta*-substituents are located in the hydrophobic environment and sterically dislocated the remaining part of the compound is bound in the hydrophilic concave of CA XII active site (Figs. 4A–4D). The favorable changes of the binding enthalpies of the dissimilar binders could be explained by the favorable location of the hydrophilic *para*-group which is sterically dislocate into the hydrophilic part of the active site where the formation of additional hydrogen bonds is more likely. The favorable $\Delta H_{interactions}$ may be the reason for the improved binding affinities.

It is interesting to compare the compounds **12** (pair **D4**) and **16** (pair **D3**) that differ by only one bridged methyl group in the *meta*-substituent. The difference in the binding energy between these compounds (**12** vs **16**) is significant ($\Delta\Delta G = 1.0 \pm 1$, $\Delta\Delta H = 17.2$, $-\Delta T\Delta S = -16.2$ kJ/mol). Maybe this could be explained by the possibility of only one position of compound **12** *para*-group when bound in CA XII, where two hydrogen bonds are formed with the residues Asn64 and Pro200. The *para*-group of **16** was found in the alternate conformation, but in the compound **12** bound to CA XII both substituent groups were found in a single conformation. Another explanation can be the same as for *similar* binders, where the increase of the hydrophobic surface induces similar behavior of binding thermodynamics. Compound **12** and **16** bind to CA XII in identical mode.

### The cases of perfect geometry fit in the interactions between protein and ligand

An extremely high affinity of the compound **1** to CA II and CA XII may be an example of a perfect geometrical fit of the ligand to the CA active site. The compound **1** has a relatively simple structure and its binding is enthalpy-driven (Fig. 4F, Table 3) exhibiting very high intrinsic affinity to both analyzed CA isoforms, 7.4 pM to CA II and 67 pM to CA XII. As seen in both crystal structures shown in Fig. 4F, the ligand is likely to bind so strongly due to the tight aromatic-aliphatic interaction between the benzene ring and a conservative leucine side chain (Leu198 in CA II and Leu197 in CA XII). An important difference between CA II and CA XII is the substitution of Phe131 in CA II with the alanine side chain in CA XII. A loss of the exothermic enthalpy was observed for the binding of **1** to CA XII as compared to CA II due to the possible absence of additional interaction of *para*-substituent of **1** with Phe131 in CA II.

Compounds of the pair **S4** (Fig. 4E) exhibit an identical orientation of the benzenesulfonamide ring as compound **1** (Fig. 4F) bound to CA II and CA XII. The $K_d$ values of **7** and **6** bound to CA II are 25 pM and 80 pM, respectively, and such values are similar to the binding of compound **1** to CA II and CA XII. Moreover, binding of these compounds are enthalpy-driven. The aromatic-aliphatic interaction looks like a possible reason of the favorable enthalpic gain (**7**: $\Delta H = -62.0$ kJ/mol and **6**: $\Delta H = -48.4$ kJ/mol) of which a major part is due to $\Delta H_{interactions}$.

Moreover, analysis of these pairs (**1** in CA II and CA XII and matched pair **S4**) highlights that there are contacts between the *para*-substituents and the protein. In the crystal structures of **1** in CA II and CA XII, the loss of additional surface (Ala instead of Phe) changes the location of the propyl group which is now directed perpendicularly to the plane of aromatic-aliphatic interaction between Leu198 and benzene ring bound to CA XII (Fig. 4F). The propyl group of the compound **1** bound to CA II (Fig. 4F) is located in the hydrophobic cavity (formed by Phe131 and Pro202) and possibly increases the affinity of the aromatic-aliphatic interaction due to same direction of interactions between (1) propyl group and hydrophobic cavity and (2) benzene ring of compound and leucine side chain.

The matched pair **S3** (Fig. 3C) exhibits similar behavior. Atoms of His200 side chain dislocate the fluorinated benzenesulfonamide ring of both compounds **5** and **3** and possibly increases the affinity of the aromatic-aliphatic interaction with Leu198. Compound **5** (of pair **S3**) binds to CA I with $K_d = 1.5$ pM (Table 3) as previously confirmed both by the thermal shift assay and displacement ITC (*Zubrienė et al., 2015*).

## DISCUSSION

The water molecules are highly important and poorly understood participants in protein-ligand association (*Klebe, 2015*; *Fox et al., 2017*). However, there are a number of difficulties and uncertainties in the analysis of water molecules in active sites of crystal structures. The number of water molecules observed in the active site of crystal structure strongly depends on the resolution and buffer composition resulting in buffer molecules located at the protein surface instead of water molecules, and the assignment of water-like electron
densities are often ambiguous. The number of water molecules in the active site cavity is also influenced by the alternative orientation of residues in the structure. In addition, adjacent protein molecules present due to crystal packing often stabilize water molecules at the interface and make them visible in the electron density maps.

Some of the structure-thermodynamics relationships helped us to design compounds that would exhibit selectivities in binding to some CA isoforms. Furthermore, such correlations sometimes help better understand the structural features of the inhibitors that determine the affinity of the compound to CA.

Analysis of binding data showed that most of the *similar* binders did not exhibit the increase of binding affinities in matched pairs, the additional molecular contacts between the protein and the ligand did not make the binding more favorable. The pair **S1** (compounds **10** and **11**—CA XIII) is an exception exhibiting 30-fold increase in binding affinity to CA XIII upon the addition of cyclooctyl group to **10**. This pair has an interesting feature of binding which is not present in other similar pairs. The access of water molecules to sulfonamide did not change after the increase of hydrophobic surface in pairs **S2–S9** (Fig. 3). Only in the pair **S1** there are changes in water molecules present near the sulfonamide and Zn(II). Several possibilities to displace water molecules upon compound binding are schematically drawn in Fig. 6. The dissociation of **10** from CA XIII (pair **S1**) could be described by the mechanism shown in Fig. 6A, while the dissociation of **11**—by the mechanism shown in Fig. 6B. The complicated path for water molecule access to displace the sulfonamide is a likely reason for the lowering of the measured by surface plasmon resonance method off-rate $k_{off}$ ($k_{off}$ of SPR: $7.9 \times 10^{-3}$ (**11**) (*Talibov et al., 2016*) vs $1.1 \times 10^{-2}$ (**10**)). The hydrophobic substituent creates a temporary barrier against water molecules that must penetrate into the active site during dissociation process. It could be the reason of better affinity for **11**.

Among dissimilar binders, the matched pair **D1** (Fig. 4A), where compounds **10** and **11** are bound to CA XII is interesting. The inhibitor **10** binds to CA XII 318 times weaker than **11**, the values of $K_d$ (Table 3): 35 nM for **10** and 0.11 nM for **11**. In the crystal structure compound **10** interacts with the particular part of the active site while the other part is filled with water molecules and the binding mechanism could be schematically shown as in Fig. 6A. The binding kinetics of the compound **11** to CA XII could not be measured by SPR due to an extremely low dissociation rate (*Talibov et al., 2016*, too slow dissociation, $k_{off} < 10^{-3}$ s$^{-1}$).

In the crystal structure of the complex CA XII with **11**, the *para*-group is located under the side chains of Asn64 and Lys69 and such binding mode may explain the lower $K_d$. The bulky hydrophobic group in *meta*-position dislocates the *para*-substituent of **11** under the side chains of amino acids in the active site cavity. It is possible that the process of dissociation of compound **11** could begin from the conformational changes of the large flexible hydrophobic group in the *meta*-position which can induce the escape of the *para*-group from under protein side chains. The bulky hydrophobic substituent can also serve as a temporary barrier that prevents the entry of water molecules during the process of dissociation. The proposed mechanism explaining the high affinity of **11** towards CA XII is schematically shown in Fig. 6C. Dissimilar binders showed that the hydrophobic group
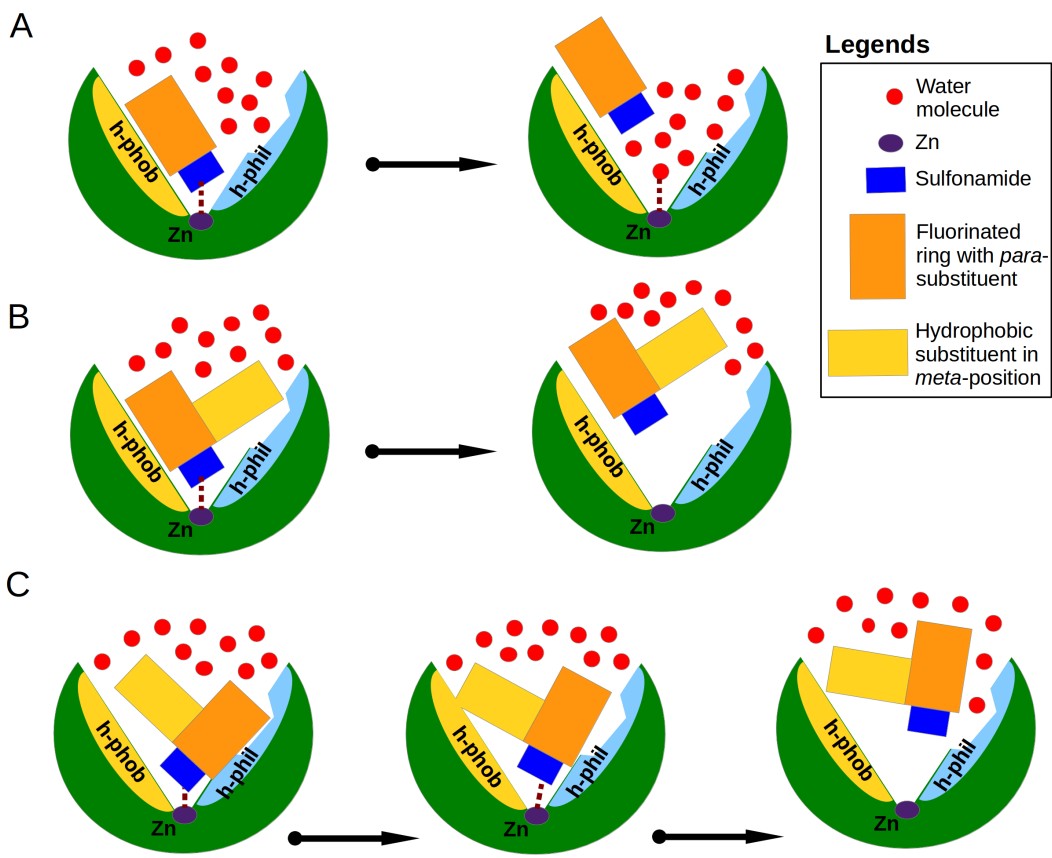

**Figure 6** **Proposed mechanisms of non-classical impact of inhibitor hydrophobic group to the binding affinity.** The CA active site is shown as green arc-shaped object where $Zn^{2+}$ ion is a small violet circle. The schematic compound consists of the sulfonamide (blue rectangle), hydrophobic substituent (yellow rectangle) and benzene ring with *para*-group (orange rectangle). The small red circles represent the water molecules. The h-phob and h-phil labels mean the hydrophobic and hydrophilic parts of schematic active site. (A) The visualization of inhibitor dissociation process when water molecules in the active site compete for the binding to sulfonamide and $Zn^{2+}$ ion and participate in the compound dissociation out of the active site. (B) A large hydrophobic group in the favorable orientation can create temporary barrier for water molecules, which cannot easy reach the sulfonamide and $Zn^{2+}$ ion. (C) The possible dual mechanism of the ligand dissociation: an additional hydrophobic group sterically dislocates the part of the ligand into the small active site cavities (molecular trap) and creates the steric barrier for water molecules to penetrate deeper into the active site.

can efficiently press the other part of the compound into small concaves of the active site and restrain the compound mobility. This opportunity may be significant for the design of the isoform-specific compounds that could distinguish between very similar active sites.

The additional molecular contacts did not make the binding affinities more favorable, but caused unfavorable changes in binding enthalpies mirrored by favorable changes in entropies. The position, direction and size of the hydrophobic group seems to play a favorable role for binding affinity only when it moves away water molecules bound deep in the active site cavity and creates a barrier for penetration of water molecules to the

ligand (Fig. 6). The complicated dissociation of the ligand away from active site can be a significant reason for higher affinity.

In this manuscript we described cases of correlation between parameters of binding thermodynamics and crystal structures of protein-ligand complexes. Some explanations and correlations could be drawn, but a systematic rational prediction of the binding thermodynamics from structure is still largely an elusive task. The dissociation mechanism of ligand from the binding site is interesting. For compounds discussed in "The cases of perfect geometry fit in the interactions between protein and ligand" we showed that the network of interactions between the ligand and the target strongly fixed the compound in the active site. For dissimilar pairs, another mechanism, where additional hydrophobic surfaces participate in ligand orientation, dislocates so that another part of the ligand is located deep in the active site and possibly complicates penetration of water molecules into the active site upon which the dissociation process has been postulated.

# CONCLUSIONS

Pairwise comparison of CA ligands enabled determining the correlations between substituent chemical structure of the ligand and the thermodynamics of binding, including Gibbs energies, enthalpies, and entropies. Compounds were allocated to two groups—similar, the ones where the orientation of the main benzene ring is the same in the absence and presence of the substituent, and dissimilar, the ones where the orientation of the main benzene ring changes upon addition of the bulky substituent. The binding of compounds comprising similar pairs was less enthalpy- and more entropy-driven upon addition of the substituent.

However, in the dissimilar pairs, the compounds bearing large substituents were more enthalpy driven than the ones where the substituents were absent. Additional hydrophobic groups enabled the localization of hydrophilic part of more hydrophobic compounds in the hydrophilic part of active site where formation of hydrogen bonds is more likely, and it can explain favorable enthalpic changes of contributions in dissimilar pairs. Moreover, the additional hydrophobic interactions can complicate the dissociation process.

# ACKNOWLEDGEMENTS

The authors thank the local contacts at the EMBL beamlines Dr. G. Bourenkov and Dr. M. Cianci for the help with P13 and P14 EMBL beamline operations at PETRAIII storage ring (DESY, Hamburg).

## Funding

This research was funded by a grant from the Research Council of Lithuania (S-MIP-17-87). The funders had no role in study design, data collection and analysis, decision to publish, or preparation of the manuscript.

## Grant Disclosures

The following grant information was disclosed by the authors:
Research Council of Lithuania: S-MIP-17-87.

## Competing Interests

Daumantas Matulis declares that he has related patent applications and patents pending.

## Author Contributions

- Alexey Smirnov conceived and designed the experiments, performed the experiments, analyzed the data, prepared figures and/or tables, authored or reviewed drafts of the paper, approved the final draft.
- Asta Zubrienė conceived and designed the experiments, performed the experiments, analyzed the data, authored or reviewed drafts of the paper, approved the final draft.
- Elena Manakova conceived and designed the experiments, performed the experiments, analyzed the data, contributed reagents/materials/analysis tools, authored or reviewed drafts of the paper, approved the final draft.
- Saulius Gražulis analyzed the data, contributed reagents/materials/analysis tools, authored or reviewed drafts of the paper, approved the final draft.
- Daumantas Matulis conceived and designed the experiments, analyzed the data, contributed reagents/materials/analysis tools, authored or reviewed drafts of the paper, approved the final draft.

## Data Availability

The raw data has been previously published and the crystal structures have been deposited to the PDB. Accession numbers: 5LLC, 5LLE, 5LLG, 5LLH, 5MSB, 5LLO and 5LLP.

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
