# Peer review of "Crystal structure correlations with the intrinsic thermodynamics of human carbonic anhydrase inhibitor binding"

_PeerJ, doi:10.7717/peerj.4412_

## Round 0.1 · original submission · Minor Revisions

Please address all the comments and requests made by the two reviewers.

Reviewer 1 ·

Basic reporting

The present manuscript analyzes the correlations between thermodynamic signature of CA inhibitor binding and the corresponding 3D-structures of the resulting CA inhibitor complexes. Pairs of protein-ligand complexes were analyzed and divided into two groups of similar and dissimilar binding poses. From this similar pairs were found to be primarily dependent on entropy changes as a consequence of desolvation thermodynamics. In contrast, in dissimilar pairs, the compounds bearing large substituents bound more enthalpy driven, which was explained by the formation of hydrogen bonds to the hydrophilic part of the active site. It was also hypthesized that additional hydrophobic groups could complicate and potentially slow down the dissociation process. The study includes 7 new crystal structures of CA-inhibitor complexes which - together with already known structures and thermodynamic parameters - form the basis for this comprehensive study of the relationship between structure and thermodynamics. The study is performed according to highest scientific standards providing a valuable and valid basis for the deeper understanding of molecular recognition. By this, the present studie extends the existing knowledge significantly.

The authors claim also consequences for the dissociation kinetics in their conclusions and even seem to intend to investigate the binding kinetics as stated in the introduction "Here we correlate the thermodynamics and kinetics of binding with the structures ...". Unfortunately, the authors do not provide kinetic data to demonstrate, for example, the expected consequences of more complicated or retarded dissociation kinetics.


minor issues:

-l209 f "...The observed entropies were obtained..." , the statement should be more precise
considering the Temperature which is multiplied with binding entropy in the underlying expression.

- Fig. 6: replace Phi- by latin letter.
The header should refer more explicitly to the mechanism of ligand dissociation.
The arrows in the figures are misleading, because they usually indicate mesomerism between
resonant chemical structures. If dissociation is meant, the arrow should have only one head.

-l500 f "... Moreover, the additional hydrophobic interactions can..." instead of
"...Moreover, the additional hydrophobic can..."

Experimental design

no comment

Validity of the findings

no comment

Additional comments

no comment

Reviewer 2 ·

Basic reporting

The present manuscript dissert over the interaction between carbonic anhydrases isoforms and ligands, by using a large dataset of crystallographic structures of such complexes, as well as thermodynamic data of such interaction from solution techniques. The dataset is vast and the possibility of interpretation is broad, and it seems that the authors have gathered the best of their evidences and effort in order to dissect and discussed extensively their ligand binding data and active-site interaction. While I fell that such work is carefully performed and brings a large set of valuable data enlightening the field of structure-ligand interaction, in particular CA, I believe that it deserves some revision in order to improve the report in the M&M, results and discussion section as follow.

Experimental design

- Title – Authors should consider stating in the manuscript “human Carbonic Anhidrase (hCA or just CA)” instead of solely CA. This will be clearer to the general reader, and not only to those searching specifically to CA.
- M&M - Authors should describe in details the experimental procedure for protein expression and purification (including construct / sequence / vector) or references for the same or origin of the protein
- M&M - Authors should describe in details the source of the ligands (synthesis, purchased, providers, or reference of the same)
- M&M - Authors should describe in details the experimental procedure for ITC and FTSA., including (but not limited to) protein /ligand / buffer components concentration, instrumental parameters, and so on
- M&M - Authors should describe with further details the crystallization experiments (sitting drop plate catalogue #, well volume, crystallization temperature) and preferably whether these solutions were from crystallization kits (please disclose the manufacturer) or lab-made (please disclose chemical manufacturer/provider and cat #) – this is important giving the well-known reproducibility problems associated to the crystallization technique.
- Results – This is a ligand-focused paper. However, the protein architecture and responses over ligand binding should be addressed in the results/discussion. Even at introduction (see line 74, “The thermodynamics of ligand binding depends on many factors, including…”, and also “(Energy) gain associated with the conformational changes of ligand and protein”) no statement is done in relation to the overall protein conformation which is observed in so many protein-ligand interactions reports. A comparative analysis of the effect of ligands on overall structure might help in understanding a broader effect of ligand binding on CA´s. ProSmart (CCP4) might assist in this context, also in understanding of the whole work since several structural details are brought to the attention and without data it becomes difficult to perform a proper appraisal of such work. There is plenty of room in supporting material. Authors are encouraged to provide (as supp material) the PDB and/or a PyMOL session which could assist the reviewer and the general reader.
- Results – line 292 “These changes may also be explained by a change in the water structure”. Please detail the meaning of “water structure”.
- Results - How is overall hydration / ASA of the whole proteins changed according to binding?
- Results – As stated in the manuscript (line 270 on), analysis of water based on crystallographic modeling of such molecules are limited by resolution and technique artifacts. Perhaps an analysis based on excluded volume could assist in further interpretation of such analysis.
- Results - “We speculate that the effects associated with the differences in the hydrophobic surface of the meta-groups have more significant impact to the binding thermodynamics than the release of water molecules from the active site by the para-benzene ring that is similar for both compounds bound to CA II.”. How do authors would consider addressing this hypothesis (themselves or others) ?
- Results – Line 349. “All molecular pairs of similar binders indicate that energies of desolvation processes significantly change the parameters of binding thermodynamics.”. Please re-elaborate this sentence.
- Results – Line 350. “The favorable changes of desolvation entropy after an increase of hydrophobic surface…”. Where the data supporting the statement in this sentence can be found ?
- Results – Line 353. “The unfavorable changes of binding enthalpy in pairs can be explained by the energy…”. Where the data supporting the statement in this sentence can be found ?
- Results – Authors claims that “The Km by both methods (FTSA and displacement ITC) agreed nearly perfectly (Zubrienė et al., 2015a). The observed entropies were obtained by subtracting the Gibbs energies obtained by FTSA from the enthalpies obtained by ITC.” Why the dG used was from FTSA and not ITC ? I´m afraid I might have missed some point or, instead, authors should address this issue accordingly.
- Results – I feel like some statements about the magnitude and order of interaction (“strengthened by aromatic-aliphatic interactions”, “formation of additional hydrogen bonds is more likely”, “explained by increase of translational and rotational mobility
- (freedom) of water molecules which were more ordered near the hydrophobic surfaces of
- compounds before desolvation”, among others) relies in the speculative realm. While hypothesis might rise from experimental work, authors should rely their interpretation preferably within the data-supported context of evidences.
- Discussion ¬– Paragraph lines 468-476. “it seems that the desolvation process is significant and may be a main reason behind the phenomenon known as enthalpy-entropy compensation.”. Authors state, with proper caution, that interpretation of water data is of limited value for a number of reasons. Should not they restrict their discussion/conclusion to this same limitation instead of using incomplete evidence from water to push some, perhaps, biased conclusion ?
- Results – throughout the manuscript authors use the expression “pushes”. There is no action being observed here. Unless they add dynamic data here (from mol dynamic simulation, Laue diffraction, apo-holo crystallographic data, NMR, among other techniques), within the context of crystallographic evidence, they are allowed to describe the “allocation” (or any other term) of the ligand in a given place/cavity, but refrain from discussion mechanical interpretation.
- Table 1 - Please include the Wilson B-factor for each structure in Table 1
- Table 1 does not include all the crystal structures associated to this manuscript. Please correct accordingly.
- Figure 1 does not include all ligands from all crystal structures reported in this manuscript. Please correct accordingly.
- Fig 3 and 4 – These figures are really a challenge to me as a reader. They contain to many elements (colors, chemical structures, 3D-graphic representation of PX data, and a bar-chart (in c panel). Too many layers, misalignment, despite the data quality it is not good from a visual design perspective. After so much effort involved in the data collection, interpretation and writing, I´m pretty confident that a final effort can be invested in a better final Figure. Just to exemplify, Fig. 4A contains four sub-panels, three in upper line and one in lower line: the four subpanels should be leaved side-by-side for the sake of comparison. All chemical structures (Fig. 2, 3, 4) have tiny fonts, which will become not-readable on a final, reduced figure upon formatting into final paper. Please consider improving this issue.
- Fig. 3 and 4 – again the water issue. Does it worth discuss the role of solvation water from crystallographic data ? Resolution matters. Crystallographic (ordered, thermodynamically stable ligand-bound) water molecules not the same as solvation (bulk, unordered) water.
- Fig 5 – authors should leave the X scale (ASA) in the same scale for both “similar and dissimilar binders” (i.e., 0 to 350 A). They are encouraged to use square panels instead of long rectangles which is currently resulting in compressed Y-axis.
- Fig 6 - authors should include in the figure a summary for their representation, independent from the legend. This would bring more prompt interpretation to the reader. Example: draw the violet circle and beside it the description “Zn”, blue rectangle next to a “sulfonamide”, and so on (mere pictorial suggestion).
-
- Manuscript - Please carefully spell-check the whole manuscript, typos and other minor issues can be found, including (but not limited to): Pigaku instead of Rigaku,

Validity of the findings

While this manuscript bring interesting results from a large dataset (crystal, solution thermodynamics), I feel not completely confident that a large presentation of the case for hydration can be inferred from such work due to several limitations already presented here by the reviewer and the authors in their manuscript. Please consider revisiting these issues, both by limiting the scope of the discussion on hydration or by introduction of further experimental (wet bench or computer simulation) evidences supporting the body of discussion throughout the manuscript.

---

## Round 0.2 · accepted · Accept

The authors have suitably corrected the manuscript according to the suggestions and comments of the reviewers.